# Membrane topography and the overestimation of protein clustering in single molecule localisation microscopy – identification and correction
Jeremy Adler [1], Kristoffer Bernhem [2] & Ingela Parmryd [1] ✉

According to single-molecule localisation microscopy almost all plasma membrane proteins are clustered. We demonstrate that clusters can arise from variations in membrane topography where the local density of a randomly distributed membrane molecule to a degree matches the variations in the local amount of membrane. Further, we demonstrate that this false clustering can be differentiated from genuine clustering by using a membrane marker to report on local variations in the amount of membrane. In dual colour live cell single molecule localisation microscopy using the membrane probe DiI alongside either the transferrin receptor or the GPI-anchored protein CD59, we found that pair correlation analysis reported both proteins and DiI as being clustered, as did its derivative pair correlation-photoactivation localisation microscopy and nearest neighbour analyses. After converting the localisations into images and using the DiI image to factor out topography variations, no CD59 clusters were visible, suggesting that the clustering reported by the other methods is an artefact. However, the TfR clusters persisted after topography variations were factored out. We demonstrate that membrane topography variations can make membrane molecules appear clustered and present a straightforward remedy suitable as the first step in the cluster analysis pipeline.

A detailed understanding of the organisation of plasma membrane components is required to understand biological processes like cell adhesion, endocytosis, host-pathogen interactions as well as the initiation of cell signalling and signal transmission. An important aspect is whether the membrane components are clustered, which covers dimerization to macroscopic assemblies, and how the extent of clustering varies with changing cellular activities. Single-molecule localisation microscopy (SMLM), including photoactivation localisation microscopy (PALM) and direct stochastic optical reconstruction microscopy (dSTORM), are often used to address these questions.

Surprisingly, SMLM suggests that nearly all plasma membrane proteins are more or less clustered, both in resting and activated cells[1]. In addition to biological considerations, there are several technical reasons to treat this ubiquitous clustering with caution; the repeated detection of single fluorophores[2], inappropriate labelling densities in relation to fluorophore on-off rates[3,4] and the failure to account for variations in membrane topography.

Repeated detection of single fluorophores has long been recognised as a potential cause of artefacts, counteracted by the use of high irradiation intensities to minimise the phenomena, the development of computational correction methods and by the use of DNA-PAINT which does not require blinking-promoting buffers[2,5–11]. Inappropriate labelling density is generally addressed using high intensity irradiation to achieve high photoswitching ratios[3].

The relevance of variations in membrane topography is generally not considered in studies of membranes and membrane processes. This is problematic since molecules that are randomly distributed in a membrane will appear to be more prevalent in areas where more membrane is present, which can be mistakenly interpreted as clustering. Importantly, it has been demonstrated that cells are neither smooth nor flat[12] and a general method to account for this in image analysis using a membrane marker has been proposed[13]. Although it has been acknowledged that membrane topography can influence the apparent distribution of molecules[14] and a method to create compensatory maps of plasma membrane undulations, but not the

[1]Department of Medical Biochemistry and Cell Biology, Institute of Biomedicine, The Sahlgrenska Academy, University of Gothenburg, Gothenburg, Sweden. [2]Science for Life Laboratory, Department of Applied Physics, Royal Institute of Technology, Stockholm, Sweden. ✉e-mail: ingela.parmryd@gu.se

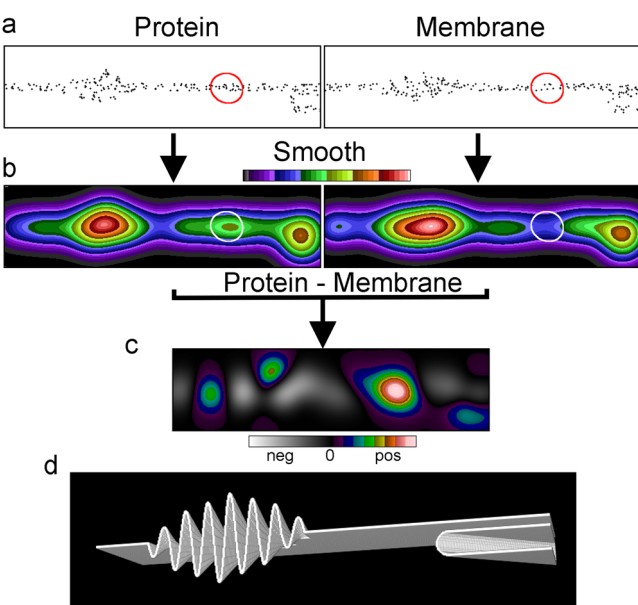

**Fig. 1 | Membrane topography variations can cause the appearance of clustering. a** Membrane cross section of a simulated SMLM dataset of a membrane protein and a randomly distributed membrane marker, 716 × 193 pixels. **b** The dataset after smoothing with a Gaussian filter (sigma 24 pixels). The images were normalised to the same mean intensity and displayed with the same intensity range. In the protein image three clusters are apparent and the least prominent and its corresponding area in the membrane image marked (white circles). The corresponding areas in the datasets in **a** are marked with red circles. **c** The difference image [protein-membrane marker] between the two normalized Gaussian filtered images. The middle and originally least prominent of the three clusters in **b** (white circle) is now seen to be genuine, whereas the two larger clusters coincided with areas rich in membrane. Note, the display has been stretched to use the full range of the look up table where zero is at the centre. **d** The rendered membrane with folds used for the simulations.

total membrane variability, has been presented[15], membrane topography variations are generally still not considered when interpreting SMLM.

Most SMLM clustering studies use fixed cells, since they permit more localisations and hence the identification of smaller clusters than live cells. However, fixation is not unproblematic. Unless rigorous fixation protocols are employed, antibodies may cause membrane protein rearrangement and clustering[16,17]. Moreover, the rigorous fixation required to avoid any rearrangement causes cells to shrink, deforming the plasma membrane[18]. The distribution of proteins also varies with temperature[16], emphasising the importance of fixation at physiological temperatures. On the other hand, in live cells molecular mobility, membrane undulations and cell movement may blur clusters unless the clusters are stable and the membrane is static. Unless the image acquisition time is appropriately fast. However, the resolution of membrane topography can improve as more localisations of the same membrane probe molecules are collected at different positions[19,20].

Clustering can be assessed using the Ripley functions[21], nearest neighbour analysis[22] and pair correlation analysis. The Ripley functions but also pair correlation analyses are prone to exaggerate the degree of clustering when the repeated localisations of the single molecules appear in the datasets[23–25]. Pair correlation-PALM was therefore introduced as a remedy to differentiate between multiple appearances and genuine clustering[5]. A common drawback with these tools is that none of them identifies the spatial distribution of the clusters, which is of biological interest.

Herein we have explored how variations in membrane topography influence the interpretation of the clustering of membrane molecules. This is illustrated by simulations and live cell dual-colour SMLM. Using intensity distribution analysis after image processing, we introduce a practical method to differentiate between genuine clusters and those that are artefacts of membrane topography variations as well as showing the distribution of clusters.

## Results

We consider the consequences of non-flat cell surfaces on clustering or, more precisely, on the appearance of clustering. Most SMLM studies on the distribution of plasma membrane proteins include the unstated assumption that the membrane is flat and therefore avoid considering membrane curvature and folding. However, neither the apical nor the basolateral side of cells are flat[12,26]. Variations in membrane topography at different scales; folds, undulations, protrusions like filopodia and invaginations like caveolae and clathrin-coated pits (CCPs), result in the membrane occupying different fractions of the local imaging volume. This alone is sufficient to make molecules that are randomly distributed appear to be clustered.

### Simulations suggest that membrane topography can cause apparent molecular clustering

To illustrate how differences in membrane topography affect the apparent clustering of a membrane protein we simulated localisations of a protein and a randomly distributed membrane marker in a variably folded membrane (Fig. 1a). Converting the localisations representing the nominal protein into an image suggests there are three clusters (Fig. 1b), whereas converting the localisations of a randomly distributed membrane marker into an image suggests there are two clusters. Normalising the mean intensities factors out differences in the number of protein and membrane marker localisations and subtracting the smoothed membrane marker image from the smoothed protein image completely changes the interpretation (Fig. 1c). Firstly, the two most prominent protein clusters almost disappear because the protein density in those regions matches that of the membrane, i.e. the clusters are not genuine. Secondly, the central and initially least prominent protein cluster not only persists but is enhanced, since here the protein density exceeds the membrane density, i.e. the cluster is genuine. A rendered membrane illustrates the underlying membrane topography with the feature on the left representing membrane ruffling and the feature on the right a collapsed filopodium (Fig. 1d).

### Established cluster analysis methods fail to recognise artefactual topography-instigated clustering

To examine how the enrichment of molecules in designated areas is reported by methods commonly used in cluster analysis, we simulated protein clusters by varying the partition ratio of single molecule localizations between designated clustered and non-clustered areas (Fig. 2). The clusters were randomly distributed and not allowed to touch. Note, the stated partition ratios are the average density within the whole area designated for clusters relative to the average density outside the clusters. Therefore, the number of localisations within individual clusters differs as they would in a stochastic process, like partitioning between membrane nanodomains or the photoactivation of fluorophores. Localisations were placed randomly both within and outside the designated cluster areas with an increasing proportion of the molecules in the cluster areas as the ratio increased. A ratio of 1 denotes the absence of intentional clusters in the image and is used to represent a membrane marker in a uniformly thick and flat membrane. Importantly, a random distribution is not a uniform distribution and hence there are areas that are either relatively enriched or relatively depleted in localisations. At low partition ratios, some areas outside the designated cluster areas have higher localisation densities than found within some of the designated clusters (Fig. 2a, Ratios 2 and 4). This is reflected in the nearest neighbour analysis where the distance to the nearest neighbour at low partition ratios for the protein are indistinguishable for the clustered protein (ClusA v ClusA) and the randomly distributed membrane marker (ClusA v Rand) neighbours (Fig. 2b). At partition ratios of 6 or more, the nearest neighbour distances between some of the proteins are noticeably shorter, coinciding with the existence of visibly detectable clusters. Pair correlation analysis, where a value of one on the y-axis indicates a random distribution and higher values indicate clustering, displays noticeable differences between the clustered and the randomly distributed molecules even at a partition ratio of 4 (Fig. 2c).

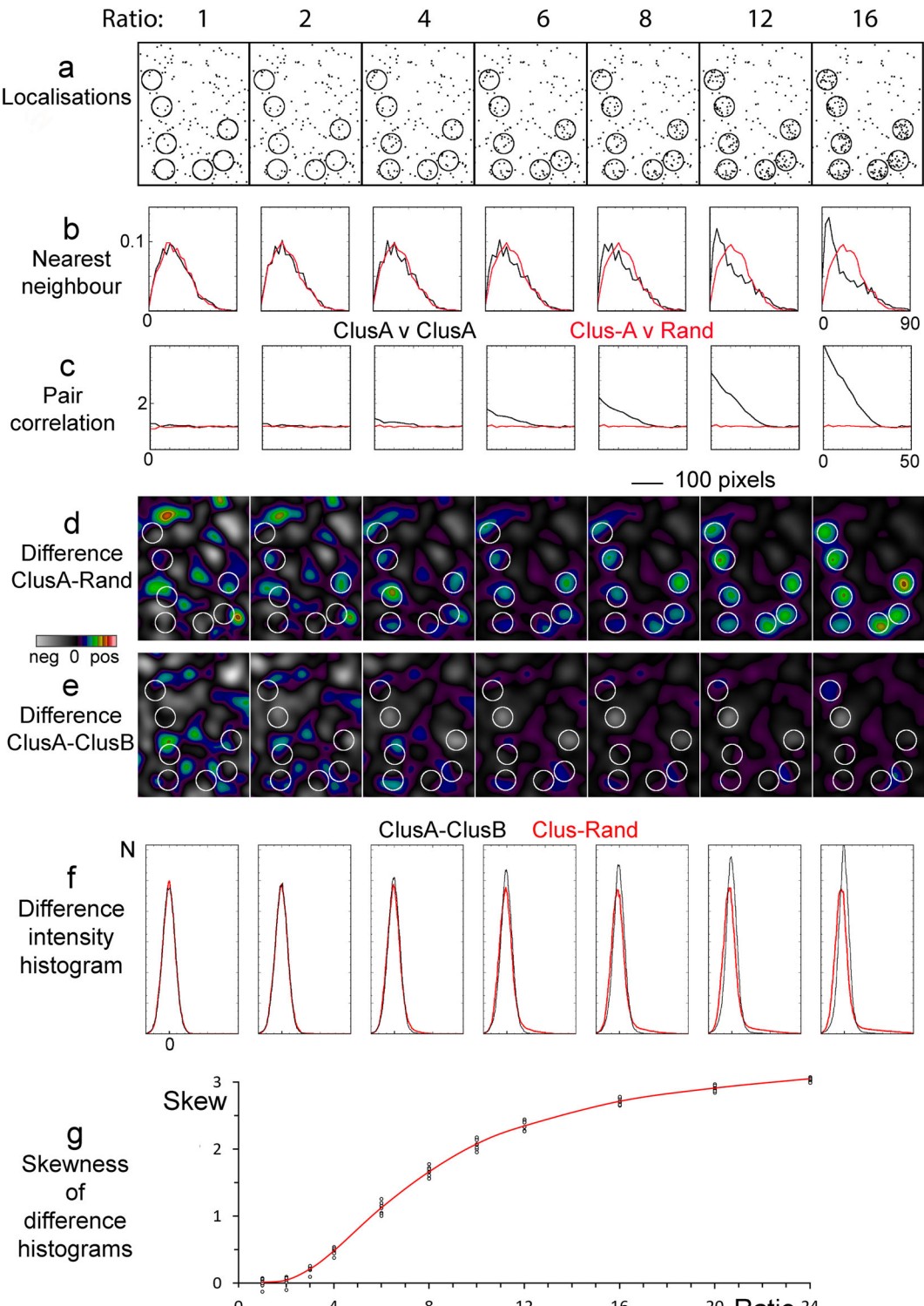

**Fig. 2 | Comparison of cluster analysis methods. a** Cluster-rich areas from larger simulated datasets (Supplementary Fig. 1). A fixed number of localizations were split between the designated cluster areas (marked, covering 4% of the total area in the full-size datasets) and non-clustered areas with incrementally increasing ratios of the localization density between the clustered (marked circles) and non-clustered areas, the total number of localizations being constant. Note the ratio of 1 means that the image does not contain areas of deliberate localisation enrichment. **b** Nearest neighbour analysis for the clustered molecule to itself and to a randomly distributed molecule. ClusA is a protein partitioning to the cluster areas at different ratios. Rand is a random distribution in a flat and smooth membrane. The scale on the x-axis represents the distance in pixels. **c** Pair correlation analysis of the clustered molecule to itself and a randomly distributed molecule. The scale on the x-axis represents the distance in pixels. **d**, **e** A cluster-rich portion from the whole 2048 × 2048 pixel dataset is displayed after Gaussian filtering (sigma 20 pixels). Subtraction of the Gaussian filtered images was performed either with (**d**) a second Gaussian filtered image of a random distribution (ClusA-Rand) or (**e**) another image of clustered molecules with the same partition ratios but different absolute distributions both inside and outside the clusters (ClusA-ClusB). ClusB is a membrane marker partitioning at the same ratios to the cluster areas as the protein marker (ClusA). For each ratio (**d**) and (**e**) use the same intensity range with zero at the centre and the maximum taken from the peak intensity in the full size image (Supplementary Fig. 1). **f** Intensity histograms from **d** (red) and **e** (black). **g** The skewness of the intensity distributions of difference image histograms, *n* = 8 with the line showing the mean. All analyses used the large datasets (Supplementary Fig. 1).

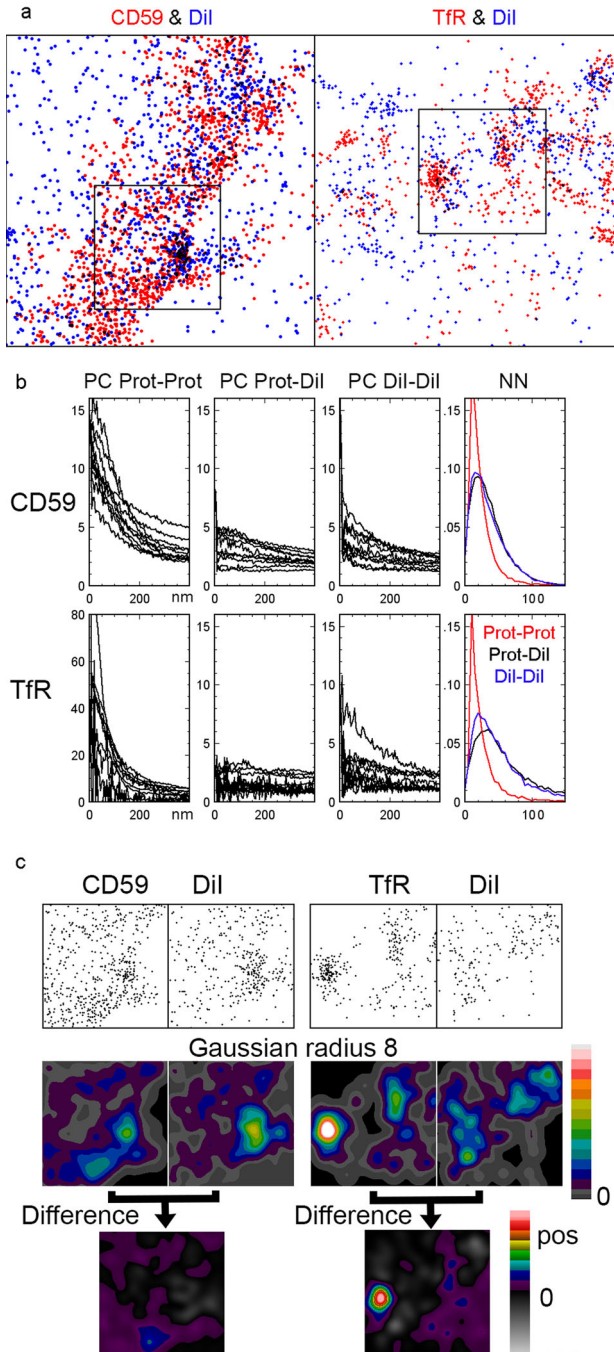

**Fig. 3 | Live cell dual colour SMLM with a protein and the membrane marker DiI to identify which clusters are caused by variations in membrane topography in HT29 cells. a** 2.5 × 2.5 µm SMLM 7 s snapshots of DiI and CD59 or the TfR with the proteins identified with Alexa Fluor-647 conjugated antibodies. Localizations of the protein are shown in red and the DiI in blue. Note areas of overlap appear black. **b** Nearest neighbour analysis and pair correlation analysis of the protein molecule to itself, to DiI and DiI to DiI. The nearest neighbour analysis is from the entire dataset. For pair correlation analysis representative results from 10 out of 25 snapshots from the entire dataset are displayed. Note that five of the six pair correlation graphs use the same y-axis range but the TfR prot-prot uses a much wider range. **c** 1 × 1 µm regions extracted from the larger images (marked with squares in **a**). Original and Gaussian filter (sigma 8 pixels) versions of all four images were normalised to the same mean intensity and displayed with the same intensity range. The final pair of images shows the result of subtraction of the DiI image from that of the relevant protein. The two Difference images use the same intensity range without any contrast alteration and the intensity range shows negative values in a greyscale.

## Intensity distribution analysis differentiate between genuine and topography-instigated clustering

To illustrate the effect of membrane topography variations on the interpretation of distributions that appear to be clustered, the simulated single molecule localisations were converted into images by smoothing with a Gaussian filter. This merges nearby localisations and shows putative clusters as areas with above average intensity. Two different images of a membrane marker were then subtracted from the image of the putative protein clusters. The first image represented a randomly distributed membrane marker in a uniformly thick and flat membrane. This resulted in visible clusters appearing in the designated cluster areas at a partition ratio of 4 or greater (Fig. 2d). However, at a partition ratio of 4 or less, putative clusters also appear outside the areas designated as clustered, a consequence of random not equating to uniform. At ratios above 4, clusters are only seen in the designated cluster areas. The second membrane marker image represents a randomly distributed marker in a membrane where the local membrane density matches the clustering partitioning ratios, i.e. any clustering here is artificial and merely follows the topography variations. Physiologically, this can occur for instance when microvilli make contact with a surface[13]. In this scenario, the putative clusters are largely eliminated by the subtraction, demonstrating that membrane topography can create the appearance of clustering, which is only exposed after factoring out the membrane topography variations (Fig. 2e). Figure 2 shows a small part of the much larger images used for the simulations and the complete smoothed images are shown in Supplementary Fig. 1.

The degree of protein clustering is reflected in the intensity variations in the difference images, which can also be visualised in intensity distribution histograms, where clusters result in skewed distributions[16]. For the difference images in Fig. 2d, e, the corresponding intensity distribution histograms are displayed in Fig. 2f. In a difference image with no designated clusters, i.e. the result of subtracting a random distribution from another random distribution, the intensity distribution is Gaussian (Fig. 2f, Ratio 1). This is despite the image being visibly inhomogeneous and appearing to contain clusters (Fig. 2d, e, Ratio 1). As the partition ratio is increased, and genuine clusters emerge in Fig. 2d, the corresponding intensity distributions have a progressively greater positive skew representing a population of high intensity pixels (Fig. 2f). There is also a small negative shift in the peak intensity caused by a lower localisation density outside clusters at higher partitioning ratios. Since the total intensities of the two images used in the subtraction is equal, the mean intensity of the difference images is zero. Quantification of the intensity distribution skewness confirmed that the skewness increased with the partitioning ratio as a random distribution was subtracted from a clustered distribution (Fig. 2g, Supplementary Data 1). Repeating the simulations ($n = 8$), the difference in skewness between the random (ratio 1) and clustered distributions was first significant at a partitioning ratio of 3 (difference = 0.20, 95% CI 0.14–0.26, $p = 1.6 \times 10^{-5}$, t = −6.71).

## Membrane topography variations can create the appearance of molecular clustering in cells

The simulations demonstrate that variations in membrane topography between pixels can create the appearance of clustering. As outlined in Fig. 1a, a remedy is to factor out any membrane topography variations using an image of a probe that distributes randomly in the plasma membrane and therefore reflects variations in topography. This enables a compensation strategy, a solution which we have previously suggested[13,27]. To this end, we employed the membrane probe DiI which is photoswitchable and expected to produce the required localisation density[19].

Accordingly, we imaged live human colon adenocarcinoma HT29 cells with combinations of DiI and either the transferrin receptor (TfR), which continuously recycles to the plasma membrane after endocytosis in CCPs, where it is enriched[28], or CD59, a GPI-anchored protein, whose clustering status together with other proteins implicated in T cell signalling is controversial[9,23,29,30]. The proteins were visualised with Alexa Fluor-647-

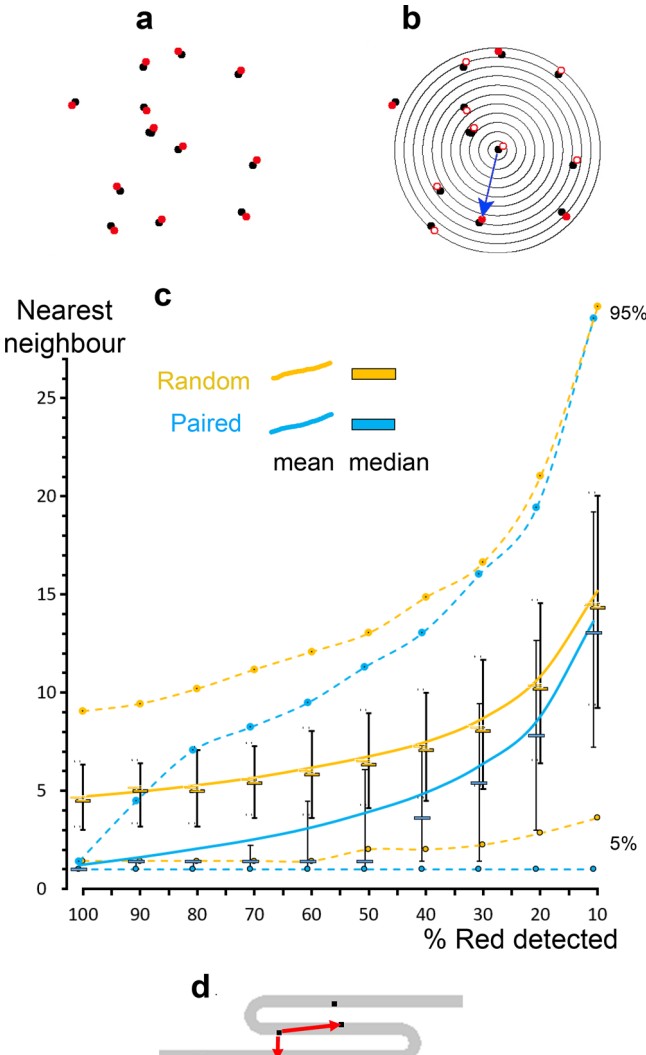

**Fig. 4 | Artefacts caused by missing molecules and membrane topography variations result in misinterpretations in nearest neighbour analysis. a** Cartoon showing randomly distributed dimers. For illustrative purposes the two molecules in each pair are displayed in different colours; red and black. **b** When some of the red molecules are not detected (empty circles), the Euclidean distance to the nearest neighbour from the black molecules increases. **c** Nearest neighbour distances as a function of the fraction of undetected molecules of randomly distributed dimers (blue) and randomly distributed molecules (orange). Whiskers display the range of the second and third quartiles. The solid lines connect the means, the median is shown with horizontal bars and the 5% and 95% values are shown as circles.
**d** Cartoon illustrating how membrane topography can cause misidentification of the nearest neighbour comparing the measured nearest neighbour with the possible nearest neighbour within a membrane or structure.

conjugated antibodies (Fig. 3a, Supplementary Data 2). We split our 25,000 frame (7 ms/frame) datasets into "snapshots" of 1000 frames, which in combination with the absence of addition of a mercapto-reagent in our buffer reduced the problem of repeated detection of molecules. Visual inspection suggested that none of the probes had a homogenous distribution. The TfR displayed substantial accumulation in areas likely to represent CCPs, which occasionally disappeared in the time-series, presumably being endocytosed, something that is expected for CCPs.

The CD59 and DiI had more even and similar distributions. Note, although imaging was performed at the basolateral cell sides there were parts of the plasma membrane that were further away from the coverslip, i.e. beyond the range excitable by total internal reflection fluorescence (TIRF) microscopy, about 150–200 nm, and hence appeared as CD59 localisation-

free zones (Fig. 3a). Outside the cell areas, there was a low density of DiI background, but no CD59.

The distributions of the three molecules were first analysed by methods that differentiate between clustered and non-clustered distributions. Pair correlation analysis suggested that all three molecules, the TfR, CD59 and DiI, were clustered (Fig. 3b). According to the pair correlation analysis TfR was more clustered than CD59 and DiI whose degree of clustering were lower and similar. Pair correlation-PALM analysis, introduced to reduce the false-clustering incidence[5], suggested that all three molecules were genuinely clustered (Supplementary Fig. 2). Nearest neighbour analysis revealed that the TfR was more likely to have another TfR molecule than a DiI molecule in its vicinity and similarly CD59 was most likely to be close to another CD59 (Fig. 3b). For DiI the outcome of the nearest neighbour analysis was less clearcut, but there was a tendency for a DiI molecule to be more likely to have another DiI molecule as its neighbour than a protein molecule (Fig. 3b and Supplementary Fig. 3).

Neither pair correlation nor nearest neighbour analysis identify the location of individual clusters and cannot establish whether the CD59 or TfR clusters coincide with DiI clusters and therefore might have arisen simply from variations in membrane topography. To this end we applied an image analysis approach - the method used with the simulated datasets. The localisations were smoothed with a Gaussian filter and the total intensities of the images normalised to compensate for differences in the number of localisations. After this, the DiI image was subtracted from the corresponding CD59 or TfR image. Using this simple approach, it was clear that the TfR was genuinely clustered while the CD59 clusters overlapped with DiI clusters and accordingly disappeared after the subtraction, i.e. the clusters simply reflected the variation in membrane density (Fig. 2c). Examining CD59/DiI and TfR/DiI datasets from two additional cells for each combination supported our initial observation that the CD59 followed variations in the membrane density whereas most of the TfR was truly clustered (Supplementary Fig. 3). Note that in the two TfR datasets displayed in Supplementary Fig. 3, single localisations will appear as low intensity circular areas when the datasets are smoothed but mostly disappear after subtraction with the DiI image. Therefore, it is important to ensure that single localisations are not identified as clusters in the difference images, which they potentially could be if the protein localisations are few and well-separated.

Next, we examined how the extent of smoothing affects the outcome of the density distribution analysis. Varying the radius of the Gaussian filter between 4-20 pixels did not alter the conclusion that the TfR is clustered, whereas CD59 followed the variations in membrane density (Supplementary Fig. 4). At small radii, this is clearer and, predictably, the TfR clusters appear larger as the radius is increased.

To test whether the low level of background DiI localisations influenced the conclusion that CD59 is not clustered, a DiI background subtraction was performed on the DiI image with the background defined as the area with no CD59 localisations. The normalisation and subtraction protocol was then repeated using the foreground area alone. The difference from the whole image subtraction (Fig. 3 and. Supplementary Fig. 4) was minimal and also after the background subtraction, the CD59 distribution visually followed that of the membrane (Supplementary Fig. 5).

## The molecular detection level and membrane topography affect nearest neighbour distances and reporting of clustering
Detection of protein interactions in dimers or larger assemblies is biologically highly relevant. However, correct assessment of the number of proteins that interact depends on the fraction of the total molecules detected and in SMLM only a fraction of the molecules are identified. Moreover, this fraction is difficult to establish and usually unknown[31]. Consider a molecule that dimerises and measured using nearest neighbour analysis. A reduction in the proportion of molecules detected increases the measured distance between the molecules and underestimates the degree of clustering (Fig. 4a–c, Supplementary Data 3). For randomly distributed molecules, the measured distances increasingly underestimate the genuine distances as the

proportion of detected molecules is decreased. As expected, the two distributions become more and more similar as the fraction of detected molecules is reduced. Note that when 90–50% of the red molecules are detected, the mean increases for the paired molecules whereas the median is constant, because in 50% or more of the pairs both molecules are detected.

Another consequence of membrane topography is that the nearest neighbour may not be in an adjacent part of the membrane, but for instance in nearby folds (Fig. 4d). Resolving membranes at this level would however require both a large number of localizations of a membrane marker and exceptional localization accuracy. Note that the intensity distribution analysis we introduce does not provide information on inter molecular distances, but is a tool to differentiate between topography-instigated and genuine clustering.

## Discussion

Fundamental biological processes like cell adhesion, signal transduction and vesicular transport occur at the plasma membrane, which is also the site of entry for pathogens. Hence, there is considerable interest in its dynamics and detailed organisation. A somewhat surprising conclusion is that the majority of the proteins examined by SMLM have been reported to form clusters[1]. We have previously shown that failing to consider variations in membrane topography causes the consistent underestimation of diffusion rates, creates the appearance of anomalous diffusion and, by altering the amount of membrane in the observation volume, complicates fluorescence correlation spectroscopy analysis[12,32,33].

We now considered the consequences of variations in membrane topography for SMLM datasets and demonstrate that this can make a random distribution appear to be clustered with the clusters appearing where the plasma membrane simply occupies a greater fraction of the imaged volume, i.e. where there is more membrane. This is possible because the plasma membrane is undulating as well as foldable and is less than 5-nm thick. Pixels in both conventional microscopy and from SMLM will therefore represent both varying amounts of plasma membrane and, correspondingly, varying amounts of membrane-associated molecules. To correctly understand the mechanisms behind biological processes at the plasma membrane, it is clearly critical to differentiate between clusters that arise from self-assembly or an interaction with other membrane components and those that are simply consequences of membrane topography variations. This requires correcting for variations in membrane topography.

TIRF–SMLM has been the dominant imaging method in super resolution cluster analysis with the underlying assumption that the basolateral side of cells is flat and therefore that a 2D dataset accurately represents the molecular distribution. However, when we imaged live adherent HT29 cells labelled with the membrane probe DiI using TIRF-dSTORM, the DiI distribution was heterogeneous, suggesting that the basolateral plasma membrane is not flat, a conclusion supported by electron microscopy[34], reflection light microscopy[35] and variable-angle TIRF[26]. However, in limited cases the plasma membrane may really be horizontal for instance in T cell activation studies where the cells adhere to an antibody-coated coverslip or to a supported bilayer containing activating and adhesion molecules, which may cause the cells to progressively flatten[36]. Whether such activation mimics the physiological interaction between two deformable cells remains to be determined, but it has been demonstrated that the interdigitation between a T cell and an antigen-presenting cell is considerable, indicating that it differs substantially from two opposed flat surfaces[37]. Moreover, it has been shown that surfaces that are generally considered inert, like polylysine, actually cause both activation of T cells and clustering of molecules involved in T cell signalling[38,39]. Interestingly, the silanized coverslips we used did not cause the GPI-protein CD59 to cluster.

In TIRF the excitation occurs within a couple of hundred nanometres of the coverslip, which can make cellular protrusions that touch the coverslip appear as clusters[13,40–42]. When fluorophores are positioned in layers at gradually increasing distances from the coverslip, the fluorescence in TIRF falls exponentially with depth[43], but in more complex samples like cells, fluorophore orientation and the varying polarisation of the excitation light makes this relationship less clear[44]. It is therefore reasonable to assume that in our TIRF-SMLM experiments, molecules within a given distance from the coverslip had the same activation probability. However, since the imaging depth is also wavelength dependent[43], it is advisable not to combine fluorophores with a large spectral difference in their emission and/or excitation. For the DiI/Alexa Fluor-647 combination we did not see signs of imaging depth mismatch, but we did observe areas of sparse labelling for the DiI/TfR and DiI/CD59 pairs indicative of distance variations between the cells and the coverslips. The distances were consistent with variable angle TIRF surface mapping showing that the distance between the cell and the surface of the coverslip is in the order of 100–200 nm for the bulk of a cell[26,45].

A consequence of acknowledging membrane topography variations in the analysis is that localizations that are coincident in time and/or space in 2D SMLM datasets may actually be separated in z and therefore should not be routinely dismissed as the repeated detection of a single fluorophore molecule, a common feature of data processing[2,31]. 3D SMLM datasets would both be helpful to determine which localisations to exclude and provide an estimate of the cluster density and volume. 3D datasets could also show whether clusters are found at specific cell surface features like invaginations or the tips of microvilli.

Importantly, 3D localizations of the molecule of interest alone are insufficient to account for membrane topography since they do not show the local amount of membrane, which is necessary to factor out clustering caused by variations in membrane topography. Hence, membrane reconstruction from either a membrane probe or an extremely abundant membrane protein is required. Assuming the topography variations only involve shallow curvature, membrane mapping can successfully be performed from xyz-stack confocal images of the basolateral cell side[46]. Using this approach, it was recently demonstrated that proteins with different shapes and properties have small but unique preferences for shallow curvatures[47]. However, topographical structures like CCPs and caveolae are not resolvable using confocal microscopy. SMLM, on the other hand, has the capacity to reveal this information. Z-localisation precision of 55 nm in live cells was achieved by SMLM pioneers using astigmatism[48]. Although this is still insufficient to resolve the detailed topography of a structure less than 5-nm thick, it would reduce any erroneous rejection of z-separated localisations. A combination of the point-spread function with biplane imaging has been reported to give a z-resolution better than 13 nm[49], but biplane imaging at this z-resolution requires a large number of photons which may be unachievable in live cell SMLM, since the reducing components that enhance blinking are toxic and incompatible with living cells. However, better blinking buffers and/or new fluorophores, could make this possible. Furthermore, high precision approaches that provide 3D position and orientation from localisations based on defocusing[50] and emission polarisation[51] may be adapted for live cell SMLM, but to achieve localisation precision in the sub nm or even the low nm scale, fixed samples are required[52,53]. Unfortunately, fixation can alter both cell shape and molecular organisation[16,54].

An additional consequence of membrane topography is that distances between membrane-associated molecules should be measured along a path that remains within the membrane, an approach we found greatly improved the accuracy of diffusion measurements[32]. Using distances that stay within the membrane is also relevant in cluster analysis methods that consider molecular densities at incrementally increasing radii, like the Ripley functions and pair correlation methods. However, on non-flat surfaces the actual distances between molecules in 2D datasets are difficult to establish but will be underestimated and these methods should therefore be treated with caution. Bias and a tendency to overestimate the degree of clustering further adds to these methods being questionable[24,55].

Another problem with nearest neighbour and pair correlation cluster analysis is that, when only a small minority of the molecules are in clusters, the majority are not and overwhelm the measurements and biologically important changes may escape notice. This is most evident in the nearest neighbour analysis where, with the simulated datasets, a higher cluster partition ratio is required for the clusters to be detected than with the pair

correlation and skewness analyses. However, both nearest neighbour and pair correlation analysis misclassify molecules that simply follow the membrane topography variations as being clustered, whereas our intensity distributions analysis using a membrane marker can expose topography-instigated clustering.

When assessing under what circumstances clustering is detectable by an algorithm, how the simulated distributions are generated will influence the outcome. In our simulations, the clusters neither touch nor overlap, which we consider physiologically relevant. Our rationale being that if the clusters originate from slight differences in lipid packing, as seen in co-existing liquid-ordered and liquid-disordered phases[56–58], the coalescence of two liquid-ordered domains will not change the fraction of the area each phase occupies. In a series of studies, a group has recently addressed the problem of overcounting by either altering the labelling density or using two different fluorophores to detect the target molecule[9,23,59]. In their cluster detection testing, the authors unlike us permit the clusters to overlap, leading to partially more prominent clusters due to denser local labelling. This approach will overestimate under what conditions clustering will be identified. Nonetheless we support their conclusion that clustering of plasma membrane components is considerably overestimated. We also think that their two-colour approach[59], if combining a membrane marker with a protein of interest, is an alternative to our intensity distribution analysis.

The basic requirement for a membrane probe able to report and thereby correct for variations in membrane topography is that it distributes randomly, both in the bulk membrane and in co-existing membrane nanodomains, without preferential partitioning. Consideration of suitable molecules suggests that even a highly expressed protein is unlikely to generate sufficient localisations to provide the necessary detail, although a general protein labelling scheme might suffice[60]. However, to establish that the protein distribution actually matches the membrane topography variation would be difficult. Therefore, a membrane probe like DiI is preferable. Although DiI in well-separated liquid disordered (ld) and liquid ordered (lo) phases, displays a preference for ld-phase[61], the composition and asymmetry of the plasma membrane are not well replicated in this type of model membrane. The difference between any co-existing phases in the plasma membrane is considerably smaller making preferential partitioning of DiI in the plasma membrane more unlikely[62]. Reassuringly we find that CD59, a GPI-anchored protein and hence a marker for plasma membrane lo-domains, did follow the membrane topography variations as mapped by DiI, consistent with DiI partitioning to both plasma membrane lo-domains and ld-domains.

An important requirement for our method is that the ratio of proteins to lipids within the plasma membrane is sufficiently stable in order to avoid the membrane probe concentration fluctuating. Reassuringly, the variation in the protein to lipid ratio across the plasma membrane appears to be minor[60]. In addition, our finding that CD59 follows membrane topography variations, are consistent with the protein to lipid ratio being similar throughout the plasma membrane.

The distributions in our simulations reflect the inherent randomness in a stochastic process like SMLM where not all molecules are detected[63,64]. An ideal membrane marker that distributed evenly rather than randomly, would provide improved mapping of the membrane but self-avoidance, though desirable, is difficult to implement. Therefore, the intensity distribution in the difference images will spread around zero for randomly distributed proteins, but our approach enables the differentiation between genuine and topography-instigated clustering. The higher the labelling density of the membrane marker the better the membrane representation, but there is a risk is that a high concentration of the probe could alter the membrane organisation, arguing against using high labelling levels. It is also important that the membrane marker is retained in the plasma membrane and does not rapidly spread to internal membranes. To minimise this risk, the imaging should take place immediately after the staining[15].

Due to the snapshot approach used to create datasets in live cell-SMLM, it is possible that small CD59 clusters remain undetected. A small cluster size reduces the probability of detecting multiple localisations in the same snapshot since these are acquired at considerably shorter times than is normal for standard datasets in fixed-cell SMLM. Small CD59 clusters may also escape discovery if rapid diffusion separates the blinking of molecules belonging to the same cluster. However, several observations argue against these two possibilities: 1) As few as three TfR molecules within a radius of about 35 nm, given the localisation precision, were sufficient to detect a cluster (Supplementary Fig. 3, left TfR dataset). The occasional CD59 cluster should then be detected if 20–40% of the GPI-anchored CD59 are, as reported, found in small clusters[65], since blinking is a stochastic process. 2) The localisation precision of CD59 was higher than that of TfR (Supplementary Fig. 5), suggesting that the TfR is more mobile than CD59 during blinking given they were labelled with the same fluorophore. I.e. if there are CD59 clusters, they do not appear to move fast on the imaging time scale. 3) The localisation precision for CD59 has a normal distribution indicative of all molecules moving to a similar extent during image acquisition (Supplementary Fig. 6). If 20–40% of GPI-proteins are found in small clusters as reported[65], this fraction is expected to have a slower diffusion than their monomeric counterpart and this should be reflected in the localisation precision.

Movement during data acquisition is a potential problem, but it has been demonstrated that the snapshot time we used results in minimal blurring due to molecular movement[66]. Moreover, at least the rough cell topography appears to be stable at timescales considerably longer than those used for our snapshots[12,47].

An important step towards a unified way to evaluate and compare cluster analysis algorithms has recently been taken[67]. However, establishing that the clusters to be characterised are genuine and not instigated by topography variations or artefactual for other reasons, is a critical step that should precede any cluster characterisation.

Our approach is suited to establish whether clustering is a topography artefact as the first step in a cluster analysis pipeline and not intended for characterisation of the details of the clustering. However, the difference images do provide information on cluster numbers and distribution; information which is not available from nearest neighbour and pair correlation analysis. Interestingly as few as three TfR localisations within a radius 35 nm was sufficient for cluster detection using our approach. We recommend that a more detailed cluster analysis is performed from the datasets where genuine clustering has been confirmed using for instance kernel density estimation or FOCAL clustering analysis[68,69]. However, it is important to bear in mind that not all protein molecules will be detected in a single live cell snapshot so the cluster size may and the number of proteins per cluster most likely will be underestimated. Naturally, our procedure can also be used for SMLM-data from fixed cells, but fixation artefacts should both be recognised and excluded.

## Conclusion

We demonstrate that variations in membrane topography can create the appearance of clustering and introduce a method for differentiating between these artefactual clusters and genuine clusters. Although we only considered SMLM, the problem and the solution apply more widely: whenever a sub-resolution structure variably occupies voxels and also around the edges of space filling structures. We argue that variations in membrane topography must be excluded to validate clustering since the existence and extent of clustering is important for the interpretation of biology.

## Methods
### Materials
Glutamine, DMEM, trypsin-EDTA and penicillin/streptomycin were obtained from GE Healthcare HyClone (Logan, UT, USA). Foetal bovine serum, enzymes, 3-aminopropyltriethoxy silane (TESPA) and chemicals were from Sigma (St. Louis, MO). Anti-CD59 Alexa Fluor-647 (clone MEM-43, Catalogue number A6-233-T025, lot 528981) and anti-TfR Alexa Fluor-647 (clone MEM-75, Catalogue number A6-235-T025, lot 528982) were from EXBIO Praha (Vestec, Czech Republic). DiI-C12 was from Thermo Fisher Scientific (Waltham, MA). HT29 human colon

adenocarcinoma cells (Catalogue number HTB-38) were from ATCC. High precision No. 1.5 coverslips were from Marienfeld (Lauda-Königshofen, Germany).

## Simulations of SMLM datasets of randomly distributed and clustered datasets with single localisations

For nearest neighbour, pair correlation and image analysis, datasets with 52 designated clusters with a radius of 32 pixels, covering 4% of the 2048 × 2048 pixel area were created. The clusters were distributed randomly with the restrictions that they did neither overlap nor extend across the edge of the image. Sequential additions of single localisations were made to areas designated as clustered from non-clustered areas as the partition ratio was altered. Each addition was individually numbered with a separate series for the clustered and non-clustered areas. This enabled the generation of related images containing different density ratios between the same clustered and non-clustered areas. Eight such clustered images were generated for each ratio.

## Cell culture

HT29 human colon adenocarcinoma cells were ATCC were cultured in DMEM supplemented with 10% foetal bovine serum, 100 U/ml penicillin, 100 mg/ml streptomycin and 2 mM glutamine. The cells were maintained at 37 °C in a humidified incubator under 5% $CO_2$. A PCR-based method was used to assess that the cell cultures tested negative for Mycoplasma infection[70].

## Cell staining

Cells were plated on silanized No. 1.5 high precision coverslips mounted in Petri dishes 34–40 h before imaging. The cells were washed twice with supplement-free DMEM and stained with 5 µg DiI-C12 in 1 ml DMEM at 37 °C for 10 min, followed by two washes with DMEM. Antibody staining was performed on ice in 50 µl DMEM for 30 min with anti-TfR diluted 50× and anti-CD59 diluted 25×, after which the cells were washed twice in DMEM without the pH-indicator phenol red.

## STORM imaging

Single molecule localization microscopy of live cells was performed at room temperature using a Carl Zeiss Elyra PS.1 microscope equipped with 405-, 488-, 561- and 642-nm activation and excitation lasers. The objective used was a Plan-Apochromate 100×/1.46 Oil (ZEISS Microscopy, Göttingen, Germany) and fluorescence was detected on a liquid-cooled iXon DU 897 EMCCD camera (Andor Technology, Belfast, UK). For STORM imaging, a multi-purpose filter set with a dichroic beam splitter reflecting 405/488/561/ 642 was used together with laser blocking filters matched to the laser lines (405/561/642) forming a 570–630 bandpass/650 longpass dual filter. Illumination was performed over an area with a 25 µm radius in the sample (1/ $e^2$ spatial irradiance distance). The stage was calibrated prior to experiments according to the manufacturer's specifications to minimize stage drift. Spectral offset was calculated from multi-spectral beads and compensated for by shifting the Alexa Fluor-647 channel localisations. Alexa Fluor-647 and DiI-C12 were initially switched into dark states by 10 s imaging with no back-pumping and 100% laser power at 642 nm and 561 nm excitation, respectively. For 2D-STORM imaging of Alexa Fluor-647, back-pumping with 0–10% (ramped linearly over the experiment) of 405 nm laser power (maximum input 10.1 mW) was used together with 100% laser power of 642 nm (maximum input power 25.4 mW). For 2D-STORM imaging of DiI-C12, back-pumping with 0–10% (ramped linearly over the experiment) of 405 nm laser power was used together with 100% laser power of 561 nm (maximum input 57.8 mW). The imaging of the two protein probes was performed on the same day at two occasions (TfR $N = 11$, CD59 $N = 7$ with 25 snapshots/dataset). The first encountered well-stained cell without signs of damage was positioned in the centre of the field of view to minimise chromatic aberration and used for data collection. This process was repeated at random locations on each coverslip until DiI internalization became apparent at which time a new sample was used. Single-molecule

fluorescence detection with the EMCCD camera was acquired with a 100 × 100 nm effective pixel size, 7 ms integration time and a gain of 100. 25,000 images were acquired for each cell in batches of 500 frames from each channel, repeated 50 times. Imaging was performed in 2 ml phenol-red free DMEM, which was supplemented with 4% glucose, 7000 units/ml of catalase and 16 units/ml of glucose oxidase to minimise phototoxic effects.

## STORM analysis of cell datasets

STORM image localization was performed using SMLocalizer[71]. The localization table was divided into STORM snapshots 1000 frames long. The mean localisation precision was 8 nm for CD59, 11 nm for DiI and 14 nm for the TfR (Supplementary Fig. 6). For each snapshot a 2.5 × 2.5 µm region of interest was selected. The complete snapshot was used for the localisation analysis, removing edge effects that otherwise occur in subsequent analyses due to lack of knowledge of the sample outside the region of interest.

## Pair correlation analysis

Each localized protein and membrane probe within the 2.5 × 2.5 µm region of interest was used as point of origin. The local density of either protein or membrane probe localisations within a 5 nm circular band of increasing radius in 5 nm increments was normalized to the total density. The analysis was performed from each protein localization:

$$g(r) = \frac{\sum_i \sum_{i \neq j} \delta((dr + r) > d_{ij} > r)}{\pi \left((r + dr)^2 - r^2\right) n \lambda} \quad (1)$$

Where $\delta$ is 1 if the particle distance falls between $r$ and $r + dr$. $n$ is the number of particles considered and $\lambda$ is the particle density (either membrane or protein localisations) within the region of interest.

In order to evaluate whether clustering was due to multiple appearances of the same fluorophore, g(r) was fitted to the equations detailed in the work by Sengupta et al.[5] in order to yield:

$$g(r)^{protein} : g(r) = \left(g(r)^{centroid} + g(r)^{protein}\right) * g(r)^{PSF} \quad (2)$$

For pair correlation analysis of simulated datasets only localisations $r_{max}$ (100 pixels) from the borders were considered as particle of origin. All points were included as possible neighbours.

## Nearest neighbour analysis

The shortest 2D Euclidean distance to a neighbouring event was determined for each protein localization within the 2.5 × 2.5 µm region of interest[22]. Nearest neighbour distances were normalized to the total number of events. For nearest neighbour analysis of simulated data only points 100 pixels from the border were considered in order to make the analysis comparable to that of pair correlation and to remove any edge effects.

## Simulations of nearest neighbour analysis

The simulations used equal numbers (4048) of red and black molecules in a 600 × 600 pixel image, with each molecule occupying one pixel. Pairs were generated by randomly distributing the black molecules and adding a red molecule, at random in one of the eight adjacent pixels. For the non-interacting population each molecule was placed randomly. Black-red nearest neighbour distances were measured from a distance map of the red molecules. Edge effects were avoided by using only the black molecules in the central 512 × 512 pixel area, around 2900 measurements. As the number of red molecules was progressively reduced, new distance maps were generated and the nearest neighbour distances remeasured. Random distribution used the ImageJ plugin RandomJ.

## Image analysis of intensity distributions

Pairs of images of the membrane marker and a protein of interest, from simulated and cell localisation datasets, were analysed by initially smoothing each dataset using a Gaussian filter. To avoid artefacts that occur at edge positions, a one-pixel-wide frame was added to the images before Gaussian

filtering. Background in the CD59-DiI datasets was defined as the area without CD59. The mean DiI background was subtracted from the smoothed DiI image and the few foreground pixels with a negative intensity value after subtraction were excluded from the foreground area. The resulting image pairs were normalised to a common intensity mean, and the membrane marker image subtracted.

## Statistics and reproducibility

Skewness was analysed for the intensity distributions of simulated datasets after image subtraction using the skewness function type 3 in the package e1071 version 1.6-8[72] in R statistical software version 3.4.3[73]. To minimize edge effects a guard area of 40 pixels was used. The null hypothesis, $\mu 1 = \mu 2$, was tested using the Welsh two-sided, two sample t-test, df = 12.749.

## Reporting summary

Further information on research design is available in the Nature Portfolio Reporting Summary linked to this article.

## Funding

## Data availability

The simulated and the SMLM datasets analysed in this study are available as Supplementary Data.

## Code availability

SMLocalizer is a freely available ImageJ plugin[71]. The ImageJ macros used to generate and analyse the simulated data are available at https://github.com/Parmryd/Membrane_Topography_and_Protein_Clustering.

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

## Acknowledgements
SMLM was performed at the Advanced Light Microscopy facility at SciLifeLab, Stockholm and we thank the staff for their support. The RandomJ plugin package was developed and made publicly available by Erik Meijering, Erasmus University Medical Center, the Netherlands. The HT29 cells was a gift of Dr. A. Blokzijl, Uppsala University, Sweden. IP discloses support for the research of this work from the Swedish Research Council (grant number 2015-04764) and Magnus Bergvall's Foundation.

## Author contributions
I.P. and J.A. conceived the study. I.P. obtained funding for the project and designed the experiments. K.B. and I.P. performed the experiments. J.A. performed the simulations and prepared the figures. All authors analysed the data. The first draft of the manuscript was written by I.P. and all authors reviewed and approved the final manuscript.

## Competing interests
The authors declare no competing interests.
