## [Peer Review File · Communications Biology]

Reviewers' comments:

Reviewer #1 (Remarks to the Author):

The manuscript by Parmryd et al. has proposed a new processing method for clustering identification based on results from single molecule localization microscopy, which could exclude clustering induced by membrane topography. This work could be a good reference for future study and the proposed method can be applied with other analytical methods to characterize clustering effectively. The scope of this work fits the journal well but several issues may need additional attention for minor revision:

1. Figure 1: Additional arrows or markers in figures are recommended to help readers understand authors' analysis. For example, the authors mentioned, "central and initially least prominent protein cluster not only persists but is enhanced..." and proper arrows or markers on relevant figures can help readers find these abovementioned clusters. Also, scale bar, central point/reference point, or x-y coordinate system is highly suggested to help readers understand changes from figure 1b to 1c better. Regarding Figure 1d, I am not sure if this scheme is necessary or not since it seems less important and also too simplified. If it was supposed to explain Figure 1b membrane marker gathering, I believe a better-quality image combined with data points shown in Figure 1a or processed shapes in Figure 1b could help readers understand it easily.
2. Figure 2b & 2c: Are these results generated based on the full-size datasets or from the selected area? In addition, it is highly suggested that the authors should clarify the meaning of ClusA, ClusB, and Rand, similar to supplement figure 1.
3. Page 9, line 198-200: The authors mentioned, "the corresponding intensity distributions have a progressively greater positive skew representing a population of high intensity pixels (Fig. 2f)". This statement may need additional consideration. According to the supplement figure 1, it can be noticed that, in ClusA-Rand group, with the ratio increasing, many pixels in positive intensities (purple, blue, green, yellow, etc.) have become black or grey but most very white pixels are still maintaining negative (from white to grey). Based on this phenomenon, a positive skewness is possibly caused by most pixel intensities tending to be more negative (black to white color), i.e. distribution mass shift to negative. On the contrary, an increasing negative skewness is caused by a major growth of positive or high intensity pixels. Therefore, it is difficult to conclude solely the skewness change is based on the growth of a small portion of high intensity pixels. Instead, the increase of area under the right tail of the intensity distribution might be a more proper parameter to indicate the pixel population in clusters.
4. Figure 3b: The authors only displayed 10 out of 25 snapshots from the entire dataset. Can these 10 snapshots represent the entire dataset? What is the selection standard?
5. Figure 3c: It is recommended that the authors should mark their selected area in Figure 3a correspondingly.
6. I understand that English may not be the native language of the authors. However, additional language check and refinement could improve the quality of this manuscript. Several grammar errors, improper wordings, and some very complicated sentence structures (mainly due to inserting many clauses in one sentence) have lowered fluency and readability. In addition, there are multiple misuses of semicolon (supposed to use a colon instead).

Reviewer #2 (Remarks to the Author):

This manuscript presents an interesting theoretical and experimental analysis of the contribution of

membrane irregularities, such as infoldings and overlapping parts to apparently detected protein clusters, such as frequently observed in high resolution fluorescence microscopy. The authors first present an in-depth theoretical analysis followed by an example of an imaging of a protein showing genuine clustering (TfR) and a protein (CD59), which correlates more strongly with the distribution of DiI suggesting, that many protein clusters detected merely reflect membrane irregularities. Furthermore, the data presents evidence, that even on the cell surface touching the substrate, TIRF-images of DiI show irregularities, suggesting that at least membranes from HT29 cells do not completely flatten out smoothly. This is an important analysis well deserving publication since it increases our awareness of potential interpretation errors due to membrane infoldings. These may lead to the fluorescence microscopical detection of protein clusters at positions of strong variations in membrane topography, even if the protein distribution in the outspread membrane is random. In future, interpretations of any fluorescence microscopical data concerning protein distributions in membranes of biological cells will have to be interpreted with caution and corrections for membrane position in space will thus be necessary or at least taken into consideration.

Overall the manuscript is well written and understandable. I only have minor suggestions:

Page 3, Line 57: The sentence: "Repeated detection is routinely minimised by the use of high irradiation intensities and repeat localisations has long been recognised as a potential cause of artefacts, resulting in the development of correction methods" This sentence needs to be rewritten for clarity.

Line 72: replace "fixations protocols" by "fixation protocols"

Page 4:, line 86; replace "Identify" by "identifies", line 88: replace "influences" by "influence"

Page 5, line 98: Introduce "at" between "neither" and "the apical.."

Line 105: replace "affects" by "affect"

Line 118 replace "filopidia" by "filopodium",

Page 6: Fig.1 line 121: I suggest to introduce "membrane cross section of" in front of "simulated... I would replace "display range" by "color code" since "display range" could also refer to the lateral extension of the rendered image

Line 131: do you mean reported or represented?

Line 135.. Note, that the different partition ratios represent (display, indicate?) the average density

Line 138: depending on the partitioning ratio?

Page 7, line 155: could you display the randomly distributed molecule in red in fig 2 a?

Lines 154 and 156: The scale on the x-axis is mentioned twice in (b) and (c) and found above (a) which is somewhat confusing. If the whole data set represents 2048 pixels in one direction the scale bar representing 100 pixels above (a) seems quite long. Later in the text it becomes clear that only a limited section of the whole data base is shown in figure 1 and the complete picture shown in the Supplement. Perhaps this could be made clearer immediately to the reader of the figure legend.

Line 162: skewness or skewedness?

Page 8, line 197: the last sentence is incomplete.

Page 9. Line 217. At this point it would be informative to mention the type of cells imaged here.

Line 261: the bottom pair of images shows...

Line 264 with negative intensities shown with a greyscale...Thus the most negative and most positive intensities are both shown in white!

Line 268 "further" or "further away"?

Line 276 "genuine clusterd" or "genuinely clustered"?

Line 310: "topography" or "topographical inhomogeneities"? line 312 insert an "of" between "some" and "the red molecules"

Page 14, line 331: introduce FCS analysis, line 337. Replace "Clustered" by "clusters"

Reviewer #3 (Remarks to the Author):

Adler et al. present a study on using correlative protein and membrane topography imaging to avoid topography-induced artifacts in the analysis of protein distributions. They first perform simulations to demonstrate the problem of ignoring topography, which they evaluate with a suite of conventional cluster analysis methods that all fail. Next, they present their solution of correlative imaging and by blurring and subtracting protein/membrane images, true clusters can be visualised and topographical effects can be eliminated. They then go on to perform live cell imaging of the membrane and the proteins TfR and CD59, showing how both proteins would appear to be clustered when analysed in isolation, but the new methodology reveals the homogenous distribution of CD59 and the clustering of TfR.

Overall the study deals with an important problem in the membrane super-resolution imaging field that is convincingly addressed. The article is well written and accessible. The introduction is comprehensive and the discussion effectively places the results in context of prior literature and highlights the importance of considering membrane topography when studying membrane protein distributions. A minor limitation of the study in its current form is the presentation and quantification of data (see comments). The topic and the methodology employed are likely to be of general interest to the readers of Communications Biology and with some minor changes the article should be acceptable for publication.

Major comments:

Quantification of data is somewhat limited. Is the experimental data all from one individual cell? If so, this is not really appropriate as acquiring data for a few cells in each condition is commonplace in SMLM. There needs to be some reporting on numbers of cell and some quantification of extent of clustering with statistics. Readers are currently presented with a single image of a few clusters that have some number attached to it, but it is difficult to interpret this.

Similarly Supp. Fig. 1 could use quantification and statistics.

The normalisation performed for the intensity distribution image analysis may be unnecessarily biased. If certain areas are illuminated differently (either due to laser pattern or illumination or TIRF angle), such as potentially in Supp. Fig. 4b you would get bias. Would it instead be possible to normalise based on an expected localisation rate for each dye, which could be calibrated elsewhere. Alternatively, normalising to total localisation number could work instead and potentially be more resistant to illumination variations.

Regarding introduction and discussion literature:

There are some further work that could be mentioned in the introduction or discussion to alert the readers to potential solutions to identified limitations. In terms of potential blinking artifacts, substantial protein imaging efforts have been undertaken with qPAINT, which under the right conditions do not suffer from the blinking problem. In general, PAINT and DNA-PAINT approaches might be beneficial in terms of avoiding blinking buffers. The authors do not mention potential phototoxicity that could be problematic. Given the size and distribution of clusters here it does not seem as though a 5 nm z resolution would be necessary to improve on the methodology and discern 3D topography. Methods such as DNA-PAINT and resPAINT for topography imaging can clearly resolve cellular features with high precision and could extend the author's methodology to 3D. Another limitation of not considering 3D imaging is that little information is collected about where clusters reside (are they in particular regions of the cell membrane?). Expansion microscopy of CD45 exclusion probably deserves a mention as well although the author's points about fixation are taken.

There is a major circular argument here that the authors need to deal with. How do we know that the live cell CD59 data does not appear unclustered because of the motion of CD59 clusters? It would be nice to see some data on fixed cells as the author's method might be able to show if fixation itself introduces clustering, although I would not expect such extra experiments to be carried out unless

trivial or the data is already there.

Minor comments:

"According to single-molecule localisation microscopy (SMLM) almost all plasma membrane proteins are clustered, a biologically unlikely finding."

This is quite a strong statement for the abstract that requires further substantiation (as is better done in main text currently).

Excellent introduction, covering relevant techniques and the current state of literature. It would however be useful nice to have some brief mention of the biological relevance of clusters. While is handled well in discussion, mentioning it in the introduction would help with motivating the importance of the work.

Fig. 2, do the simulations take blinking into account? The authors claim that blinking is reduced with photobleaching but this is not particularly convincing without more data.

The colormap needs numbers.

"We split our 25 000 frame (7ms/frame) datasets into "snapshots" of 1000 frames, reducing or even eliminating the problem with repeated detection of molecules."

This needs better substantiation, either by rereferring to photokinetics of the dyes used or by imaging dyes on a coverslip. Probes are well known to have short repeated blinks and longer on/off duration as well.

Fig. 3 shows an incredibly pronounced cluster (TfR), what is the sensitivity of your approach? What is the smallest cluster you would expect to be able to detect under experimental conditions? Please discuss.

Supp. Fig. 4 CD59 looks quite different from the membrane, is this an issue of different TIRF angles or something else? Comment please. This could also use more quantitative analysis to establish the ideal conditions.

Not convinced about how helpful Fig. 4 is. It does not quite fit with the prior analysis or arguments. If I have missed something, its usefulness should be made more clear in text with reference to prior results.

Typographical comments:

"whereas converting the localisations of a randomly distributed membrane marker into an image suggest there two clusters."

Grammar

Referee expertise:

Referee #1: Cellular biophysics, fluorescence imaging

Referee #2: Microscopy, Neurobiology

Referee #3: Super-resolution microscopy; cellular biophysics

Reviewers' comments:

Reviewer #1 (Remarks to the Author):

The manuscript by Parmryd et al. has proposed a new processing method for clustering identification based on results from single molecule localization microscopy, which could exclude clustering induced by membrane topography. This work could be a good reference for future study and the proposed method can be applied with other analytical methods to characterize clustering effectively. The scope of this work fits the journal well but several issues may need additional attention for minor revision:

1. Figure 1: Additional arrows or markers in figures are recommended to help readers understand authors' analysis. For example, the authors mentioned, "central and initially least prominent protein cluster not only persists but is enhanced..." and proper arrows or markers on relevant figures can help readers find these abovementioned clusters. Also, scale bar, central point/reference point, or x-y coordinate system is highly suggested to help readers understand changes from figure 1b to 1c better. Regarding Figure 1d, I am not sure if this scheme is necessary or not since it seems less important and also too simplified. If it was supposed to explain Figure 1b membrane marker gathering, I believe a better-quality image combined with data points shown in Figure 1a or processed shapes in Figure 1b could help readers understand it easily.

We have added further annotation to make it clearer that panel c is the results of subtracting the "membrane" from the "protein" image in panel b. The area that becomes prominent after subtraction is now outlined with circles in panels a and b.

Panel d is a cartoon that illustrates the 3D topography that was the basis of the 2D simulated datasets, which has now been clarified. We do not think that superimposing the two localisation datasets from panel a on the cartoon would add to the understanding.

2. Figure 2b & 2c: Are these results generated based on the full-size datasets or from the selected area? In addition, it is highly suggested that the authors should clarify the meaning of ClusA, ClusB, and Rand, similar to supplement figure 1.

All analyses were performed from the full-size datasets, which has now been clarified in the figure legend as has the meaning of ClusA, ClusB and Rand. In addition, the text describing the methods has been expanded.

3. Page 9, line 198-200: The authors mentioned, “the corresponding intensity distributions have a progressively greater positive skew representing a population of high intensity pixels (Fig. 2f)”. This statement may need additional consideration. According to the supplement figure 1, it can be noticed that, in ClusA-Rand group, with the ratio increasing, many pixels in positive intensities (purple, blue, green, yellow, etc.) have become black or grey but most very white pixels are still maintaining negative (from white to grey). Based on this phenomenon, a positive skewness is possibly caused by most pixel intensities tending to be more negative (black to white color), i.e. distribution mass shift to negative. On the contrary, an increasing negative skewness is caused by a major growth of positive or high intensity pixels. Therefore, it is difficult to conclude solely the skewness change is based on the growth of a small portion of high intensity pixels. Instead, the increase of area under the right tail of the intensity distribution might be a more proper parameter to indicate the pixel population in clusters.

Subtracting an image from another image with the same mean intensity results in a mean of zero. As a consequence, there will be more pixels with negative values to balance the appearance of pixels with high intensity values. This we state “There is also a negative shift in the peak intensity caused by a lower localisation density outside clusters as the partitioning ratio increases...” The last part of this sentence has been rewritten for increased clarity.

If you have a skewness value of 0 that does not necessarily mean that you have a symmetrical distribution because it can, as you are hinting at, be the result of the combination of one long thin tail and one short fat tail. When you have a positive skew, however, this is the outcome of a dominating tail at the positive end. If anything, the existence of a simultaneous short fat tail at the negative end would decrease the skewness value.

Furthermore, even if the skewness value to some extent would underestimate the level of clustering due to a counteracting tail on the negative side, we still believe it is a better measurement than the area under the tail on the positive side since the same area could be found for a short fat tail and a long thin tail although only the latter is indicative of clustering.

4. Figure 3b: The authors only displayed 10 out of 25 snapshots from the entire dataset. Can these 10 snapshots represent the entire dataset? What is the selection standard?

Yes, we believe they can. The reason for displaying the data from 10 rather than 25 snapshots is that 25 curves would be very difficult to differentiate. The 10 snapshots were selected to represent the span seen for the different markers for the pair correlation analyses. That the snapshots are representative has now been clarified in the manuscript.

5. Figure 3c: It is recommended that the authors should mark their selected area in Figure 3a correspondingly.

The two area have now been marked.

6. I understand that English may not be the native language of the authors. However, additional language check and refinement could improve the quality of this manuscript. Several grammar errors, improper wordings, and some very complicated sentence structures

(mainly due to inserting many clauses in one sentence) have lowered fluency and readability. In addition, there are multiple misuses of semicolon (supposed to use a colon instead).

We find this a very common comment when the author affiliations is not in a country where English is the native language. As it happens, English actually is the native language of one of the authors. However, we take your point about long sentences and have broken up several of them. We have also made clarifications and corrected mistakes pointed out by the other reviewers.

Reviewer #2 (Remarks to the Author):

This manuscript presents an interesting theoretical and experimental analysis of the contribution of membrane irregularities, such as infoldings and overlapping parts to apparently detected protein clusters, such as frequently observed in high resolution fluorescence microscopy. The authors first present an in-depth theoretical analysis followed by an example of an imaging of a protein showing genuine clustering (TfR) and a protein (CD59), which correlates more strongly with the distribution of DiI suggesting, that many protein clusters detected merely reflect membrane irregularities. Furthermore, the data presents evidence, that even on the cell surface touching the substrate, TIRF-images of DiI show irregularities, suggesting that at least membranes from HT29 cells do not completely flatten out smoothly. This is an important analysis well deserving publication since it increases our awareness of potential interpretation errors due to membrane infoldings. These may lead to the fluorescence microscopical detection of protein clusters at positions of strong variations in membrane topography, even if the protein distribution in the outspread membrane is random. In future, interpretations of any fluorescence microscopical data concerning protein distributions in membranes of biological cells will have to be interpreted with caution and corrections for membrane position in space will thus be necessary or at least taken into consideration.

Overall the manuscript is well written and understandable. I only have minor suggestions:

Thank you for the careful reading. In the absence of a comment below, we have changed the text according to your suggestions.

Page 3, Line 57: The sentence: “Repeated detection is routinely minimised by the use of high irradiation intensities and repeat localisations has long been recognised as a potential cause of artefacts, resulting in the development of correction methods” This sentence needs to be rewritten for clarity.

Line 72: replace “fixations protocols” by “fixation protocols”

Page 4:, line 86; replace “Identify” by “identifies”, line 88: replace “influences” by “influence”

Page 5, line 98: Introduce “at” between “neither” and “the apical..”

Line 105: replace “affects” by “affect”

Line 118 replace “filopidia” by “filopodium”,

Page 6: Fig.1 line 121: I suggest to introduce “membrane cross section of” in front of “simulated...”

I would replace “display range” by “color code” since “display range” could also refer to the lateral extension of the rendered image

We now use the term intensity range.

Line 131: do you mean reported or represented?

Reported is correct.

Line 135.. Note, that the different partition ratios represent (display, indicate?) the average density

This sentence has been rewritten.

Line 138: depending on the partitioning ratio?

This sentence has been rewritten.

Page 7, line 155: could you display the randomly distributed molecule in red in fig 2 a?

This has been addressed in the figure legend in order not to cause confusion with the use of red in the rest of this Figure 2 as well as Figure S3.

Lines 154 and 156: The scale on the x-axis is mentioned twice in (b) and (c) and found above (a) which is somewhat confusing. If the whole data set represents 2048 pixels in one direction the scale bar representing 100 pixels above (a) seems quite long. Later in the text it becomes clear that only a limited section of the whole data base is shown in figure 1 and the complete picture shown in the Supplement. Perhaps this could be made clearer immediately to the reader of the figure legend.

We find it clearer to state what the scale bar represents for both panel b and c and verify that the size of the scale bar in panel a is correct. We are now referring to Supplementary Figure 1 earlier in the figure legend.

Line 162: skewness or skewedness?

Skewness is the correct word.

Page 8, line 197: the last sentence is incomplete.

Page 9. Line 217. At this point it would be informative to mention the type of cells imaged here.

Line 261: the bottom pair of images shows...

Line 264 with negative intensities shown with a greyscale...Thus the most negative and most positive intensities are both shown in white!

There are no pixels with high negative values in this figure. We take your point though and have changed the colour of the highest intensity pixels from white to pink in all figures.

Line 268 “further” or “further away”?

Further away.

Line 276 “genuine clusterd” or “genuinely clustered”?

Line 310: “topography” or “topographical inhomogeneities”?

This sentence has been rewritten.

Line 312 insert an “of” between “some” and “the red molecules”

Page 14, line 331: introduce FCS analysis, line 337. Replace “Clustered” by “clusters”

This sentence has been rewritten.

Reviewer #3 (Remarks to the Author):

Adler et al. present a study on using correlative protein and membrane topography imaging to avoid topography-induced artifacts in the analysis of protein distributions. They first perform simulations to demonstrate the problem of ignoring topography, which they evaluate with a suite of conventional cluster analysis methods that all fail. Next, they present their solution of correlative imaging and by blurring and subtracting protein/membrane images, true clusters can be visualised and topographical effects can be eliminated. They then go on to perform live cell imaging of the membrane and the proteins TfR and CD59, showing how both proteins would appear to be clustered when analysed in isolation, but the new methodology reveals the homogenous distribution of CD59 and the clustering of TfR.

Overall the study deals with an important problem in the membrane super-resolution imaging field that is convincingly addressed. The article is well written and accessible. The introduction is comprehensive and the discussion effectively places the results in context of prior literature and highlights the importance of considering membrane topography when studying membrane protein distributions. A minor limitation of the study in its current form is the presentation and quantification of data (see comments). The topic and the methodology employed are likely to be of general interest to the readers of Communications Biology and with some minor changes the article should be acceptable for publication.

Major comments:

Quantification of data is somewhat limited. Is the experimental data all from one individual cell? If so, this is not really appropriate as acquiring data for a few cells in each condition is commonplace in SMLM. There needs to be some reporting on numbers of cell and some quantification of extent of clustering with statistics. Readers are currently presented with a single image of a few clusters that have some number attached to it, but it is difficult to interpret this.

Here there must be some misunderstanding. As stated in the Materials and Methods, 11 cells with the TfR and 7 with CD59 were imaged. From those, one of each is displayed and analysed in Figure 3 and an additional two of each in Figure S3. We find this sufficient for proof of principle. The text has been altered to make this clearer.

Our method is not intended to quantify cluster statistics but as a first step in the pipeline to decide whether such quantification is meaningful. This we had laid out in the discussion “Our approach is better suited to establish whether clustering is a topography artefact than to characterise the details of the clustering.” We further suggested that “A more detailed cluster analysis could be performed using kernel density estimation or FOCAL clustering analysis.” We have now clarified that a more detailed cluster analysis can be performed from the subtraction images generated by our approach.

Similarly Supp. Fig. 1 could use quantification and statistics.

It is somewhat unclear what you mean here. Figure S1 shows the complete dataset of the simulated clusters with the small area shown in Figure 2 marked. We have extended the number of simulations and performed statistical analysis of the skewness values.

The normalisation performed for the intensity distribution image analysis may be unnecessarily biased. If certain areas are illuminated differently (either due to laser pattern or illumination or TIRF angle), such as potentially in Supp. Fig. 4b you would get bias. Would it instead be possible to normalise based on an expected localisation rate for each dye, which could be calibrated elsewhere. Alternatively, normalising to total localisation number could work instead and potentially be more resistant to illumination variations.

We agree that inhomogeneous illumination deserves much more attention when images are analysed and can corrupt measurements. To minimize problems with uneven illumination, imaging was performed on cells at the centre in the field of view, which is now stated in the text. Any residual inhomogeneity in the illumination will have a similar effect on both fluorophores and hence not unduly bias the difference images.

As we mention in the discussion, we could see no signs of a differential imaging depth for the two fluorophores used, DiI and Alexa Fluor-647, which do not have a large difference in their excitation and emission peaks.

The localisation rate for a dye inserted in the cell plasma membrane or attached to a membrane protein may differ from that of the same dye on its own. It is therefore not trivial to extrapolate numbers on the expected localisation rate. Importantly, our point is that the membrane topography varies and where less membrane is close to the coverslip there will be fewer localisations. We therefore do not think comparisons to the expected localisation rates will add anything to the analysis.

We are converting the localisations to images by blurring them with a Gaussian filter and normalise the intensities of the pair of images. So we are in effect normalising to the number of localisations. This has now been clarified in the text. We also examined using Voronoi tessellations, but this proved to be less satisfactory method of converting localizations in to images.

Regarding introduction and discussion literature:

There are some further work that could be mentioned in the introduction or discussion to alert the readers to potential solutions to identified limitations. In terms of potential blinking artifacts, substantial protein imaging efforts have been undertaken with qPAINT, which under the right conditions do not suffer from the blinking problem. In general, PAINT and DNA-PAINT approaches might be beneficial in terms of avoiding blinking buffers.

Thank you for the suggestions. Fixation is required for DNA-PAINT and res-PAINT, methods as you point out, have advantages over the more conventional PALM and STORM regarding blinking artefacts. Methods to perform molecular level quantification of PAINT-data are now mentioned in the introduction. Please note that our main point is not about blinking, but to increase awareness of and address the problem caused by topography variations.

The authors do not mention potential phototoxicity that could be problematic.

This is why we included the enzymes glucose oxidase and catalase in the imaging buffer. This is now explained in the text.

Given the size and distribution of clusters here it does not seem as though a 5 nm z resolution would be necessary to improve on the methodology and discern 3D topography. Methods such as DNA-PAINT and resPAINT for topography imaging can clearly resolve cellular features with high precision and could extend the author's methodology to 3D.

When you have a folded membrane, for instance a collapsed filopodium as outlined in Figure 1d, the folds are in close proximity of one another. Obtaining the 3D topography as the cell outline would miss such folds as well as invaginations like caveolae and clathrin coated pits. Since membrane folds (see Fig 4b) and also the necks of invaginations produce misleading results on molecular vicinity, the ability to resolve the membrane would be advantageous. The resolution of DNA-PAINT and resPAINT is not sufficient to differentiate between adjacent membranes like those seen for instance of collapsed filopodia illustrated in Figure 1d.

Another limitation of not considering 3D imaging is that little information is collected about where clusters reside (are they in particular regions of the cell membrane?).

We do not quite understand what you are referring to. We do discuss why 3D imaging was not possible, but agree that it would be better than 2D imaging for characterising clusters. Importantly, our method is not intended for such characterisation but as a first step in the pipeline to decide whether such quantification is meaningful. To determine whether the distribution of a membrane molecule merely follows that of the membrane 2D imaging is sufficient. Furthermore, information of where any clusters reside can be obtained since their spatial distribution is obtained and it can be determined where in the cell membrane they are positioned.

Expansion microscopy of CD45 exclusion probably deserves a mention as well although the author's points about fixation are taken.

Expansion microscopy is a fascinating method, but it precludes establishing the membrane topography with fluorescent membrane markers. Instead, the positions of membrane proteins are mapped after extensive fixation. We are not convinced that the two can be equated even for abundant proteins like CD45.

There is a major circular argument here that the authors need to deal with. How do we know that the live cell CD59 data does not appear unclustered because of the motion of CD59 clusters? It would be nice to see some data on fixed cells as the author's method might be able to show if fixation itself introduces clustering, although I would not expect such extra experiments to be carried out unless trivial or the data is already there.

We agree that it would be interesting to follow fixation and see whether it induces clustering. We currently not have any data on fixed cells and think this will be better suited for a study on that topic alone. However, we do not think it would answer the question raised— if no clusters are seen after fixation the case for them not existing would be stronger, but if they are

seen after fixation you could still argue that it is uncertain whether they were moving too fast to be detected in the live cells or are the result of fixation. Given the potential fixation artefacts, we think it is a strength and not a weakness that our study is performed on live cells.

If there was diffusion of CD59 clusters in the bulk membrane you would expect

1) The diffusion of these clusters to be faster than that of clathrin coated pits where the TfR accumulates before it is internalised. Interestingly, in our experiments the localisation precision of CD59 was higher than that of the TfR, suggesting that CD59 moves less than the TfR during blinking given that both proteins were labelled with the same fluorophore.

2) CD59 clusters to diffuse slower than monomeric CD59 giving rise two peaks on the localisation precision distribution. Importantly, the localisation precision distribution for CD59 was normal suggesting that there is one population, not two with different diffusion times. This is not supportive of a 20-40 % fraction of the CD59 being found in small clusters as suggested by some previous studies.

3) To see the occasional cluster since most molecules do not move far from their starting point but follow the probability distribution function. From the examination of the TfR datasets, as little as three localisations within a 20-25 nm radius would be identified as a cluster.

The discussion of course boils down to the definition of what is a cluster, but anything smaller than three molecules may not meet the criteria of being a cluster.

So from what we see, there does not appear to be any genuine CD59 clusters. We take your point though and have extended the discussion accordingly to include much of the reasoning above. In addition, we already mentioned that small clusters with correspondingly few localisations may be missed.

Minor comments:

“According to single-molecule localisation microscopy (SMLM) almost all plasma membrane proteins are clustered, a biologically unlikely finding.”

This is quite a strong statement for the abstract that requires further substantiation (as is better done in main text currently).

We have removed the second part of the sentence, but would like to keep the first part that is well backed-up by the literature.

Excellent introduction, covering relevant techniques and the current state of literature. It would however be useful nice to have some brief mention of the biological relevance of clusters. While is handled well in discussion, mentioning it in the introduction would help with motivating the importance of the work.

We agree and are now mentioning the biological relevance of clusters in the introduction.

Fig. 2, do the simulations take blinking into account? The authors claim that blinking is reduced with photobleaching but this is not particularly convincing without more data. The colormap needs numbers.

No, these simulations do not take blinking into account and for proof of principle that is not necessary. We wanted to keep the simulations relatively simple and they demonstrate how variation of the amount of membrane in a pixel's catchment area will make a randomly distributed membrane component appear clustered if this variation is not taken into account in the analysis. Blinking is another problem and not the focus of our study.

The look up table has a zero in the centre and to guide the readers we have now marked what is positive and negative in relation to the zero. The scale is linear but the actual numbers are arbitrary, making a scale with numbers superfluous.

“We split our 25 000 frame (7ms/frame) datasets into “snapshots” of 1000 frames, reducing or even eliminating the problem with repeated detection of molecules.”

This needs better substantiation, either by rereferring to photokinetics of the dyes used or by imaging dyes on a coverslip. Probes are well known to have short repeated blinks and longer on/off duration as well.

Thanks for pointing this out, the phrasing was unclear. What we were intending to refer to was that the live cell imaging, in combination with the use of snapshots, reduces the problem with repeated detection. Since the presence of blinking promoting molecules like mercaptoethylamine is not compatible with living cells, blinking in live cell SMLM is less prominent than in fixed cell SMLM. As a consequence, there is also less repeated detection. We have, however, not performed any photokinetic studies of the dyes in our imaging buffer and cannot present any data on this. The statement has been rewritten for clarification.

Please note that the pattern of the TfR clusters we observe is not the typical pattern seen upon reimaging the same fluorophore. Moreover, if reimaging of the same fluorophore would have been the reason why TfR clusters were seen, the expectation would be a similar pattern for CD59 since imaging was performed under identical conditions and the two proteins labelled with the same probe.

Fig. 3 shows an incredibly pronounced cluster (TfR), what is the sensitivity of your approach? What is the smallest cluster you would expect to be able to detect under experimental conditions? Please discuss.

Thank you for this question. It prompted us to look into this in more detail and we conclude that a cluster as small as three detected TfRs within a radius of 35 nm from one another is sufficient to detect as a cluster using our approach. This example is taken from the TfR datasets to the left in Figure S3. Please note that since we are doing live cell imaging divided into snapshots, far from all molecules of any type are visualised in each snapshot so characterisation of the absolute number of molecules in a cluster is not possible.

Supp. Fig. 4 CD59 looks quite different from the membrane, is this an issue of different TIRF angles or something else? Comment please. This could also use more quantitative analysis to establish the ideal conditions.

Figure S4 is a further exploration of the data from the cell used in Figure 3. We do not agree that the CD59 and DiI datasets are quite different. Given the stochastic process of imaging

and the physical impossibility of two molecules being in the exact same place in combination with the localization error, identical datasets are not to be expected. As we lay out in the discussion, we could see no signs of a differential imaging depth for the two fluorophores used, DiI and Alexa Fluor-647. As is clear from the difference images after normalizing the images, the intensities are close to zero and no clustering is seen.

The main message of Supplementary Figure 4 is that if there is clustering, you will see this using a large range of sigmas for the added Gaussian filter. For further characterisation of the clusters in the difference image, a good starting point is to use a Gaussian filter that matches the localisation precision. In this way, the cluster sizes can be estimated. This is now mentioned in the discussion.

Not convinced about how helpful Fig. 4 is. It does not quite fit with the prior analysis or arguments. If I have missed something, its usefulness should be made more clear in text with reference to prior results.

In the context of SMLM, the points raised in Fig. 4 are relevant. In Figures 2-3 we are using Nearest Neighbour analysis to assess clustering and in Figures 1-3 we are demonstrating how topography variations can lead to the erroneous conclusion that there are clusters. We have now emphasised the relevance to clustering and topography further.

Typographical comments:

“whereas converting the localisations of a randomly distributed membrane marker into an image suggest there two clusters.”

Grammar

This mistake has been corrected.

REVIEWERS' COMMENTS:

Reviewer #1 (Remarks to the Author):

The authors have sufficiently addressed my previous concerns and comments. I have no further comments regarding this manuscript.

Reviewer #2 (Remarks to the Author):

This manuscript has been considerably improved.
I just recommend to delete s in "datasets" on line 125 in the caption of figure 1.

Reviewer #3 (Remarks to the Author):

See PDF

The authors have made some effort to deal with my comments. Overall I find their response highly defensive and often unnecessarily so as all I have asked for is further quantification of images, where no quantification is there to be found, and further discussion points. Hopefully the authors are now able to take my comments on board after further clarification. Nevertheless, this is a useful methodology highlighting an important problem of topography effects in studying membrane protein distributions.

Author response in blue and reviewer comments in black

Here there must be some misunderstanding. As stated in the Materials and Methods, 11 cells with the TfR and 7 with CD59 were imaged. From those, one of each is displayed and analysed in Figure 3 and an additional two of each in Figure S3. We find this sufficient for proof of principle. The text has been altered to make this clearer.

The authors claim that their method can clearly identify real clusters by accounting for membrane topography, yet they are hesitant to analyse their data, except for direct visualisation. Any convincing visualisation can easily be quantified and tested. With data from 11 and 7 cells for each protein respectively, I do not understand why this is not analysed as part of Fig. 3. The figure does not mention data from multiple cells and the readers should not have to dig through materials and methods to get an idea of significance. Looking at Fig. 3c it seems straightforward to extract some relative information about clustering (number of peaks, max values, median peak intensity) with some kind of thresholding. With minimal effort and minimal claims of interpreting meaning other than the ability to differentiate, the authors would be able to demonstrate the ability of their methodology to discern clusters. Currently readers are presented with a selected area that could very well be biased. I have no doubt that this is representative, but I do not see any reason to not quantify the data.

It is somewhat unclear what you mean here. Figure S1 shows the complete dataset of the simulated clusters with the small area shown in Figure 2 marked. We have extended the number of simulations and performed statistical analysis of the skewness values.

See previous comment. Now that the authors have shown that skewness can be used to discriminate clusters in the simulations, why not apply that to the experimental data?

We are converting the localisations to images by blurring them with a Gaussian filter and normalise the intensities of the pair of images. So we are in effect normalising to the number of localisations. This has now been clarified in the text. We also examined using Voronoi tessellations, but this proved to be less satisfactory method of converting localizations in to images.

I remain unconvinced by the author's argument. Imagine a protein that in one image happens to have many small clusters and one enormous cluster. The normalisation using mean intensity (generated from localisations) will greatly reduce the relative intensity of the small clusters. In another field of view without an enormous cluster, suddenly these small clusters may appear quite prominent. Unless I have misunderstood something, this is an inherent bias that needs to be resolved to allow the technique to output meaningful data. Perhaps normalising to a widefield image (prior to super-resolution imaging) might be more suitable. Alternatively, as I suggested, using the expected localisation rate for each dye (which should be a fixed value given a known laser power, and could be determined using a number of ways). You mention

that the rate is likely to vary depending on dye environment, but this is a problem you would have in your blurred images as well.

When you have a folded membrane, for instance a collapsed filopodium as outlined in Figure 1d, the folds are in close proximity of one another. Obtaining the 3D topography as the cell outline would miss such folds as well as invaginations like caveolae and clathrin coated pits. Since membrane folds (see Fig 4b) and also the necks of invaginations produce misleading results on molecular vicinity, the ability to resolve the membrane would be advantageous. The resolution of DNA-PAINT and resPAINT is not sufficient to differentiate between adjacent membranes like those seen for instance of collapsed filopodia illustrated in Figure 1d.

Are the authors claiming that they achieve 5 nm resolution required to image membrane folds but DNA-PAINT, or even MINFLUX, would not be capable of performing this in 3D? I find this statement unconvincing without any evidence provided, I certainly do not observe any folds in the authors' data. In order for the authors to achieve this adjacent membrane discrimination, the cells would have to be two-dimensional, which given the importance of topography addressed in this manuscript is clearly not the case. I am quite surprised by the suggestion that the added information obtained by 3D imaging is somehow worse than pure 2D imaging (3D astigmatism does not significantly reduce 2D precision).

We do not quite understand what you are referring to. We do discuss why 3D imaging was not possible, but agree that it would be better than 2D imaging for characterising cluste. Our method is not intended for such characterisation but as a first step in the pipeline to decide whether such quantification is meaningful. To determine whether the distribution of a membrane molecule merely follows that of the membrane 2D imaging is sufficient. Furthermore, information of where any clusters reside can be obtained since their spatial distribution is obtained and it can be determined where in the cell membrane they are positioned.

The authors discuss why 3D imaging was not possible but it would be helpful to discuss what 3D imaging might add in the future. Your method of imaging the topography informs about membrane density and potential curvature, but little information is gained about the type of structure. 3D imaging would for example show if you are imaging the tip of a membrane protrusion/fold/hole etc. My suggestion is only that this limitation and potential future avenue is mentioned in the discussion.

Expansion microscopy is a fascinating method, but it precludes establishing the membrane topography with fluorescent membrane markers. Instead, the positions of membrane proteins are mapped after extensive fixation. We are not convinced that the two can be equated even for abundant proteins like CD45.

While it has not been done yet, there is no reason why correlative imaging of protein and membrane cannot be imaged with expansion microscopy e.g. [10.1021/acsnano.1c11015](https://doi.org/10.1021/acsnano.1c11015). I remain convinced that expansion microscopy would represent a potential avenue to further the arguments made by the authors in the future.

Figure S4 is a further exploration of the data from the cell used in Figure 3. We do not agree that the CD59 and DiI datasets are quite different. Given the stochastic process of imaging

I have gone through the trouble of highlighting exactly what I mean as the difference is very obvious (especially after the colour correction I have applied). Potentially this could be background noise. I hope you now understand why I also suggested TIRF angle as a possible culprit as imaging different heights of the membrane would manifest in what is observed here.

59 CD59

Membrane

A similar effect can also be seen for TfR now that the authors have made me have a second look. Perhaps this is because TfR is clustered, but who knows?

TfR

Membrane-TfR
Sign

Reviewer #1 (Remarks to the Author):

The authors have sufficiently addressed my previous concerns and comments. I have no further comments regarding this manuscript.

Reviewer #2 (Remarks to the Author):

This manuscript has been considerably improved.
I just recommend to delete s in “datasets” on line 125 in the caption of figure 1.

Thank you, the s in datasets in this and the subsequent sentence has been deleted.

Blue = selected parts of our original response

Black= new comments from reviewer #3

Red= our response to the new comments

Reviewer #3

The authors have made some effort to deal with my comments. Overall I find their response highly defensive and often unnecessarily so as all I have asked for is further quantification of images, where no quantification is there to be found, and further discussion points. Hopefully the authors are now able to take my comments on board after further clarification.

Nevertheless, this is a useful methodology highlighting an important problem of topography effects in studying membrane protein distributions. **Author response in blue** and reviewer comments in black.

We are sorry that you found some of our answers highly defensive. Mostly we were trying to clarify misunderstandings but it is clear that we disagree about the purpose of the study. We want to start a discussion in the community about how topography can create the appearance of clustering of molecules in membranes and propose a method to factor this out.

1. Here there must be some misunderstanding. As stated in the Materials and Methods, 11 cells with the TfR and 7 with CD59 were imaged. From those, one of each is displayed and analysed in Figure 3 and an additional two of each in Figure S3. We find this sufficient for proof of principle. The text has been altered to make this clearer.

The authors claim that their method can clearly identify real clusters by accounting for membrane topography, yet they are hesitant to analyse their data, except for direct visualisation. Any convincing visualisation can easily be quantified and tested. With data from 11 and 7 cells for each protein respectively, I do not understand why this is not analysed as part of Fig. 3. The figure does not mention data from multiple cells and the readers should not have to dig through materials and methods to get an idea of significance. Looking at Fig. 3c it seems straightforward to extract some relative information about clustering (number of peaks, max values, median peak intensity) with some kind of thresholding. With minimal effort and minimal claims of interpreting meaning other than the ability to differentiate, the authors would be able to demonstrate the ability of their methodology to discern clusters.

Currently readers are presented with a selected area that could very well be biased. I have no doubt that this is representative, but I do not see any reason to not quantify the data.

Our method is not intended to quantify cluster statistics but as a first step in a pipeline to decide whether any subsequent quantification is meaningful. We therefore find the difference images of CD59-DiI, where the appearance of clustering disappears upon subtraction, much more revealing than those of TfR-DiI, where clustering persists after subtraction. To characterize the clusters, we refer the readers to some of the methods already available to this end. This has now been furthered clarified in the discussion.

It is stated that data is presented from 2x3 cell datasets (3xCD59 and 3xTfR) in the Results section.

In Figure 3, a smaller area from the datasets is used for the difference images for illustrative purposes. However, larger areas from the same datasets are displayed in Supplementary Figure 4 and it is clear that the same conclusion can be drawn from both areas – that the distribution of CD59 follows the distribution of the membrane marker DiI, whereas that of the TfR does not.

2. It is somewhat unclear what you mean here. Figure S1 shows the complete dataset of the simulated clusters with the small area shown in Figure 2 marked. We have extended the number of simulations and performed statistical analysis of the skewness values.

See previous comment. Now that the authors have shown that skewness can be used to discriminate clusters in the simulations, why not apply that to the experimental data?

Originally, you asked for quantification of the data in Figure S1 which we have provided by demonstrating that skewness measurements can be used to estimate if there is clustering in an image.

The purpose of our study is to alert the community to a potential problem and how it can be addressed using a straightforward and simple approach – factoring out variations in membrane topography. This we have done by flagging an unrecognised problem and providing a workable solution.

3. We are converting the localisations to images by blurring them with a Gaussian filter and normalise the intensities of the pair of images. So we are in effect normalising to the number of localisations. This has now been clarified in the text. We also examined using Voronoi tessellations, but this proved to be less satisfactory method of converting localizations in to images.

I remain unconvinced by the author's argument. Imagine a protein that in one image happens to have many small clusters and one enormous cluster. The normalisation using mean intensity (generated from localisations) will greatly reduce the relative intensity of the small clusters. In another field of view without an enormous cluster, suddenly these small clusters may appear quite prominent. Unless I have misunderstood something, this is an inherent bias that needs to be resolved to allow the technique to output meaningful data. Perhaps normalising to a widefield image (prior to super-resolution imaging) might be more suitable.

Alternatively, as I suggested, using the expected localisation rate for each dye (which should be a fixed value given a known laser power, and could be determined using a number of ways). You mention that the rate is likely to vary depending on dye environment, but this is a problem you would have in your blurred images as well.

In both scenarios that you describe, many small clusters and the combination of a large cluster and many small clusters, our approach would flag the distributions as clustered if the clustering is not due to variations in membrane topography.

Our recommendation is that when clustering is flagged, the original datasets are analysed using existing cluster analysis methods that are based on the distribution of the localisations to characterise the clusters. This has now been further clarified in the discussion. Whether those methods would identify all clusters in a mixture of tiny and large clusters is likely but not certain – it very much depends on the distributions and the fractions of the molecules that are not in clusters. The methods would, however, undoubtedly define these kinds of distributions as clustered. Importantly, our method is not intended to quantify cluster statistics but as a first step in a pipeline to decide whether such quantification is meaningful.

Your thoughtful examples have made us put in a warning regarding sparse single localisations in the results section to preempt the unlikely event that someone might identify these as clusters.

4. When you have a folded membrane, for instance a collapsed filopodium as outlined in Figure 1d, the folds are in close proximity of one another. Obtaining the 3D topography as the cell outline would miss such folds as well as invaginations like caveolae and clathrin coated pits. Since membrane folds (see Fig 4b) and also the necks of invaginations produce misleading results on molecular vicinity, the ability to resolve the membrane would be advantageous. The resolution of DNA-PAINT and resPAINT is not sufficient to differentiate between adjacent membranes like those seen for instance of collapsed filopodia illustrated in Figure 1d.

Are the authors claiming that they achieve 5 nm resolution required to image membrane folds but DNA-PAINT, or even MINFLUX, would not be capable of performing this in 3D? I find this statement unconvincing without any evidence provided, I certainly do not observe any folds in the authors' data. In order for the authors to achieve this adjacent membrane discrimination, the cells would have to be two-dimensional, which given the importance of topography addressed in this manuscript is clearly not the case. I am quite surprised by the suggestion that the added information obtained by 3D imaging is somehow worse than pure 2D imaging (3D astigmatism does not significantly reduce 2D precision).

Here there seems to be a misunderstanding. Nowhere do we claim 5 nm resolution in SMLM. You may be referring to membrane folds but given the size of the extracellular glycocalyx, membrane folds are unlikely to result in tight membrane apposition meaning that a such high resolution may not be necessary to resolve membrane folds. It definitely would require 3D imaging though and we are also not saying that 2D imaging, which is what we have used, can be used to resolve 3D structures. The discussion on 3D imaging has been modified for clarification.

What we do observe are variations in membrane topography, which affect the apparent distribution of membrane molecules. The resolution required to move from the detection of more membrane to establishing the local topography that caused it is substantial and clearly desirable but, importantly, not required for our approach to work. We do not know the detailed membrane topography, only where there is more membrane and this alone is sufficient to discriminate molecules that follow the membrane topography variations from those that are genuinely clustered.

5. We do not quite understand what you are referring to. We do discuss why 3D imaging was not possible, but agree that it would be better than 2D imaging for characterising clusters. Our method is not intended for such characterisation but as a first step in the pipeline to decide whether such quantification is meaningful. To determine whether the distribution of a membrane molecule merely follows that of the membrane 2D imaging is sufficient. Furthermore, information of where any clusters reside can be obtained since their spatial distribution is obtained and it can be determined where in the cell membrane they are positioned.

The authors discuss why 3D imaging was not possible but it would be helpful to discuss what 3D imaging might add in the future. Your method of imaging the topography informs about membrane density and potential curvature, but little information is gained about the type of structure. 3D imaging would for example show if you are imaging the tip of a membrane protrusion/fold/hole etc. My suggestion is only that this limitation and potential future avenue is mentioned in the discussion.

We agree that it is important to elucidate the detailed topography from both protein distribution and physiological process perspectives. It is also of importance for thorough cluster characterisation but, importantly, our approach determines whether a protein distribution is merely following that of membrane marker our method and not intended to quantify cluster statistics. For this the 2D approach is sufficient as illustrated in our study.

We do discuss several aspects of 3D imaging since we agree that it would be advantageous and have made some changes to make this clearer.

6. Expansion microscopy is a fascinating method, but it precludes establishing the membrane topography with fluorescent membrane markers. Instead, the positions of membrane proteins are mapped after extensive fixation. We are not convinced that the two can be equated even for abundant proteins like CD45.

While it has not been done yet, there is no reason why correlative imaging of protein and membrane cannot be imaged with expansion microscopy e.g. [10.1021/acsnano.1c11015](https://doi.org/10.1021/acsnano.1c11015). I remain convinced that expansion microscopy would represent a potential avenue to further the arguments made by the authors in the future.

You may be correct, but it still remains to be determined how the membrane topography is preserved upon the extensive sample preparation required for expansion microscopy. However, our study is on live cell SMLM and we find a discussion on expansion microscopy too peripheral to the topic and have chosen not to include it.

7. Figure S4 is a further exploration of the data from the cell used in Figure 3. We do not agree that the CD59 and DiI datasets are quite different. Given the stochastic process of imaging

I have gone through the trouble of highlighting exactly what I mean as the difference is very obvious (especially after the colour correction I have applied). Potentially this could be background noise. I hope you now understand why I also suggested TIRF angle as a possible culprit as imaging different heights of the membrane would manifest in what is observed here.

According to the calibrations, which were performed according to the instructions of the manufacturers, there was no mismatch in the imaging depths of wavelengths used. We therefore think that your other suggestion that there is some DiI background is correct. We have therefore returned to our data and tested how a background subtraction would affect our conclusions. We find that subtracting a constant background mean, as expected, makes little difference. Please see the Supplementary Figure 5. If anything, CD59 appears to be even less clustered after the background subtraction. This is an important validation of our approach and we thank you for your persistence.

A similar effect can also be seen for TfR now that the authors have made me have a second look. Perhaps this is because TfR is clustered, but who knows?

For a clustered protein, like the TfR, it is expected that the cell surface areas between the clusters has little or at least much less protein. It is therefore more difficult to define the foreground using the distribution of the protein to identify the background where there is no membrane.

We do not think there is any doubt that the TfR is clustered since it clearly does not follow the distribution of the membrane marker DiI. A background subtraction would not alter this conclusion and we recommend that cluster characterisation is performed from the SMLM datasets rather than the subtraction images.

Clustering of the TfR is furthered supported by the use of the same fluorophore, Alexa-647, to visualize both CD59 and the TfR under identical imaging conditions, making repeated detection of the same fluorophore an unlikely explanation for the TfR clusters.

You end with “Perhaps this is because TfR is clustered, but who knows?”

After correcting for topography, the point of the study, we now have a much better chance of finding out.